# Performance Assessment of Three Turbulence Models Validated through an Experimental Wave Flume under Different Scenarios of Wave Generation

**Lander Galera-Calero** [1] **, Jesús María Blanco** [1,2,* ] **, Urko Izquierdo** [1]
**and Gustavo Adolfo Esteban** [1]

1   Department of Energy Engineering, Faculty of Engineering of Bilbao-UPV/EHU, Plaza Ingeniero Torres
    Quevedo, 1, CP 48013 Bilbao, Bizkaia, Spain; lander.galera@ehu.es (L.G.-C.); urko.izquierdo@ehu.es (U.I.);
    gustavo.esteban@ehu.es (G.A.E.)
2   TECNALIA, Basque Research and Technology Alliance (BRTA), Astondo Bidea, Edificio 700, 48160 Derio,
    Bizkaia, Spain
*   Correspondence: jesusmaria.blanco@ehu.es; Tel.: +34-94601-4250

**Abstract:** This study aimed to adjust the turbulence models to the real behavior of the numerical wave flume (NWF) and the future research that will be carried out on it, according to the turbulence model that best adjusts to each particular case study. The $k$-$\varepsilon$, $k$-$\omega$ and large-eddy simulation (LES) models, using the volume of fluid (VOF) method, were analyzed and compared respectively. The wavemaker theory was followed to faithfully reproduce the waves, which were measured in an experimental wave flume (EWF) and compared with the theory to validate each turbulence model. Besides, reflection was measured with the Mansard and Funke method, which has shown promising results when studying one of the most critical turbulent behaviors in the wave flume, called the breaking of the waves. The free surface displacement obtained with each turbulence model was compared with the recorded signals located at three points of the experimental wave flume, in the time domain of each run, respectively. Finally, the calculated reflection coefficients and the amplitudes of the reflected waves were compared, aiming to have a better understanding of the wave reflection process at the extinction zone. The research showed good agreement between all the experimental signals and the numerical outcomes for all the turbulence models analyzed.

**Keywords:** CFD; experimental wave flume; numerical wave flume; reflection; turbulence modelling; VOF

## 1. Introduction

Computational modelling is one of the most important tools when working in offshore and coastal engineering applications. Its increasing interest can be seen in academia where an important number of articles have been published in relation to wave generation [1], wave breaking [2], floating wind platforms behavior [3], or wave energy converters performance [4]. More specifically, computational fluid dynamics (CFD) show, supported by the evolution of the hardware, great potential in the study of these topics. This is reflected in the creation of numerical wave flumes (NWFs) [5] and numerical wave tanks (NWTs) [6] in the last years. They are used to support the research in experimental infrastructures but not to substitute them, because CFD applications still cannot replace the experimental work [7].

In most of these studies, the turbulence appears as a vital part of the process. Because of this, a good definition is of a great importance when trying to reduce the total cost of testing prototypes [8]. Thus, the correct selection of the turbulence model is essential in the different methods, such as the Eulerian multiphase volume of fraction (VOF) [9] or the smoothed particle hydrodynamics method

(SPH) [10], when studying these types of processes. Nowadays, the most extended turbulence models in engineering applications are the two-equation models *k-ε* and *k-ω* [11]. Both methods are applied to the Reynolds average Navier–Stokes (RANS) equations to have a better definition of the turbulence. They add two transport equations, where the first one is the turbulence kinetic energy (*k*) and the second defines the turbulence dissipation (*ε*) and the specific dissipation rate (*ω*). Both models have been developed and modified all over the years, with the realizable *k-ε* [12] and the *k-ω* Menter's shear stress transport (SST) [13] being the most known ones due to their improvements on the original models.

The offshore and coastal research has used these two models extensively in the last years for solving diverse technological challenges. The *k-ε* model developed by Launder and Sharma [14] is the most used model throughout the different engineering applications [8]. In ocean and coastal research, the *k-ε* model has been used to study the propagation of solitary waves and their interaction with submerged barriers [15] and submerged slotted barriers [16]. In these studies, Wu and Hsiao studied the free surface displacement, velocity, and vorticity generated by the submerged barriers, as well as the reflection, dissipation, and transport coefficients. Propagation of group of waves in shallow waters has been also studied [17] besides the study of the wave breaking. This last area of research has been studied by several researchers, such as Xie [18], who applied the *k-ε* model for spilling and plunging breaking waves. Moreover, Bakhtyar [19] studied these types of breaking waves and the evolution of waves in a sloping bed, along with the velocity field and its vertical distribution. Casalone et al. [7] used this turbulence model to study the motion of a hull.

The other two-equation model, the *k-ω*, which was developed by Wilcox [20] and afterwards modified by Menter [13], has also been used in offshore and coastal applications due to its better prediction of near-wall turbulence. Devolder [21] applied a buoyancy term in the model in order to avoid wave damping in the propagation of the waves. Once this term was applied, the model was used to study regular wave run-ups around a monopile. With the same model and buoyancy term, Devolder also studied wave breaking [22]. With the application of the buoyancy term, the turbulence model significantly reduced the overestimation of the turbulent kinetic energy in the flow field. Higuera et al. [23] used the turbulence model for realistic wave generation by combining the theories of piston-type wave maker motion and active wave absorption, which is also used for the study of sediment transport. Jacobsen compared the numerical model with experimental data to study the cross-shore sediment transportation, both free surface displacement and turbulence, over a fixed breaker bar [24,25]. Oggiano [26] used the model to reproduce extreme events by simulating long crested irregular waves around a monopile in 2-D. The *k-ω* was also used by Zhou et al. to study the response of a floating offshore wind turbine (FOWT) for a first-order irregular waves group [3]. Besides, some researchers have compared both methods in order to define their differences and see how the selection of each one affects to the final result [27].

However, another turbulence model has arisen in the last years and has shown promising results in the field of marine energies. This model is named large-eddy simulation (LES) and it was firstly introduced by Deardoff [28] for engineering applications. Hieu et al. used the LES model with the VOF method to study propagation, shoaling, breaking, and reflection of waves [2]. Lakehal and Liovic [29] used this model to focus on the wave breaking of steep water waves on a constant slope. Besides, Lubin et al. [30] used the model to study plunging breaking waves. Finally, Thorimbert et al. used the strengths of this model to simulate the behavior of oscillating water column devices (OWC) [31].

The scope of this paper was to directly compare different turbulence models for future research, such as OWC device behavior or breaking waves studies. The tuning of a turbulent model capable of reproducing such a high dissipative process as wave breaking will lead to wide possibilities in modeling wave–structure interaction by means of CFD codes in the marine energy field. This study allows us to see the behavior of three turbulence models generating second-order waves with a moving wall and the energy dissipation of an extinction system. Besides, an optimization of the computational mesh was done to reduce the computational cost. In Section 2, the experimental wave flume (EWF) and its instrumentation is defined, as well as the NWF and the models used. Then, the methodology

followed is defined in Section 3. The turbulence models are presented in Section 4, and then, the results are presented and discussed in Section 5. Finally, in Section 6, some conclusions are presented.

## 2. Description of the Wave Flume

### 2.1. Experimental Wave Flume

All the simulations are replicas of the tests carried out in the EWF of the Energy Engineering Department at the School of Engineering in Bilbao (University of the Basque Country, UPV/EHU). The EWF is 12.5 m long, 0.6 m wide, and 0.7 m high. The walls of the flume are made of laminated and tempered glass and the tank base of stainless steel. The commercial software ASDA-Soft (V5) [32] controls the piston-type wavemaker of the flume by controlling the servo drive and servo motor. The period and amplitude of the paddle are defined in the software, which converts the input data in a sinusoidal curve of progressive acceleration and deceleration of the motor. The motor is connected to a linear actuator, which is attached to the paddle, creating the linear movement of the paddle. Then, the intended wave is generated according to the wavemaker theory [33,34] for a piston-type wavemaker. The wave energy extinction area consists of a self-designed parabolic extinction system, which can be adjusted in height and slope to situate it in the position of maximum wave absorption. The scheme of the extinction system can be seen in Figure 1.

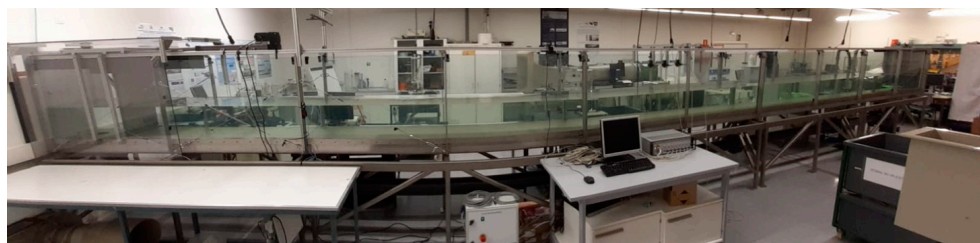

**(a)**

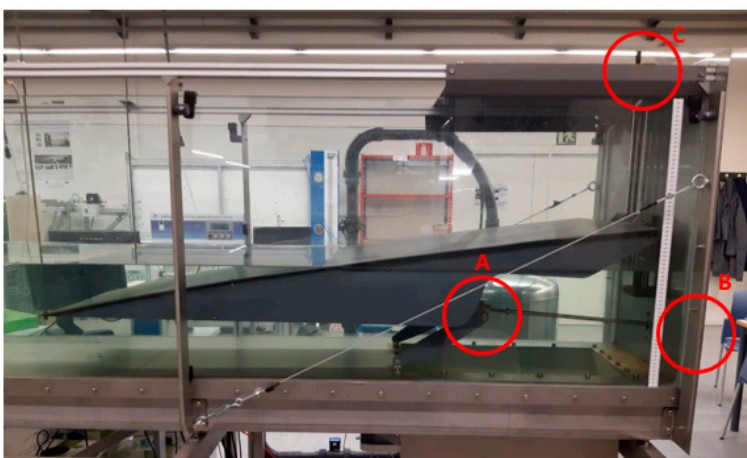

**(b)**

**Figure 1.** (**a**) The general view of the EWF from the piston-type wavemaker (left) to the extinction system (right). (**b**) Picture of the self-designed extinction system installed in the wave flume and the highlights of the join (A) and the position of the cranks (B and C) to modify the angle

The free surface variation ($\eta$ (m)) is measured in three different points by using resistive-type probes, the position of which can be modified to adjust them according to the wavelengths of the waves generated. The free surface displacement is recorded by means of an ad hoc LABVIEW in time intervals of 3 ms. It is important to highlight that the probes must be separated in equal distances, as it can be observed in Figure 2, avoiding the half and quarter of the wavelength of study to avoid errors due to their positioning. The first resistive probe is situated at 5.5 m from the paddle to ensure, at least, a minimum distance of two wavelengths of the wave generated. In this study, the wavelengths of all the waves are smaller than 2.75 m, which allow, on the one hand, the waves to be fully developed when reaching the first probe and, on the other hand, to have a maximum flume length for each study.

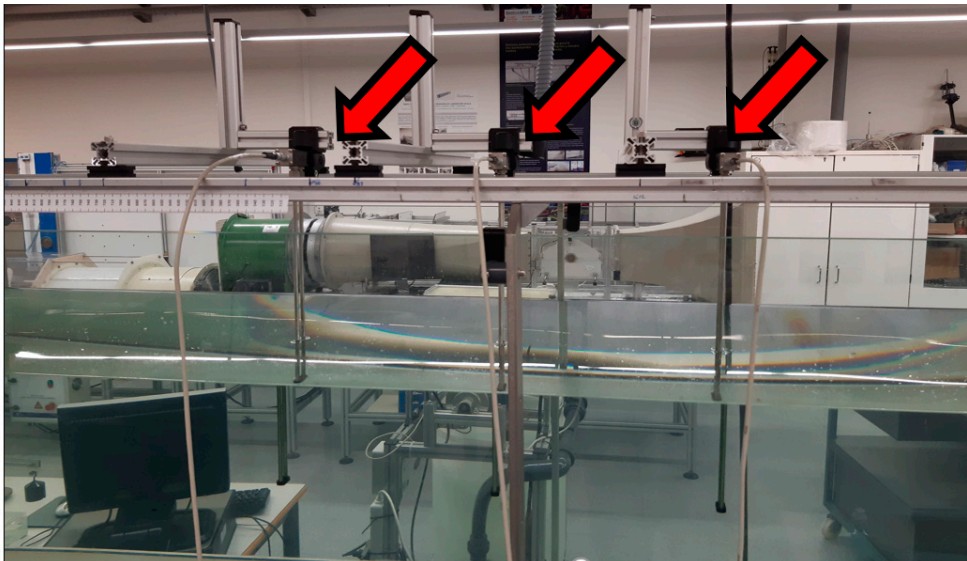

**Figure 2.** Disposition of the resistive probes in the EWF. The arrows highlight the join with the structure that allows them to move along the flume.

*2.2. Numerical Wave Flume*

The NWF, unlike the EWF, has a 12-m length, due to the elimination of the volume of water that is behind the wavemaker. As this volume has no influence in the wave generation and reflection, its implementation is unnecessary and would only increase the computational cost of each simulation.

2.2.1. The Mesh

For the two-equation models, a 2-D mesh is generated, while for the LES simulations, a 3-D one is used. In both meshes, the paddle is created as a moving wall in one end of the flume, which follows first- or second-order sinusoidal curves, depending on the wave planned. The condition introduced to the wall in the simulations is a grid velocity, so that the equations must define the velocity of the wall in each time-step. Then, the first-order equation is:

$$v(t) = (2A \cdot \pi/T) \cdot \sin(2\pi \, t/T), \tag{1}$$

where $A$[m] is the amplitude of the paddle, $T$[s] the period, and $t$[s] the time. For the second-order curve, the equation of velocity is:

$$v(t) = [(2A_1/T)\pi \cdot \sin(2\pi/T \cdot t)] - [(4A_2/T)\pi \cdot \sin(4\pi/T \cdot t)], \tag{2}$$

where $A_1$ [m] is the amplitude of the principal curve and $A_2$ [m] the amplitude of the secondary one. The meshes can be divided in different areas. The air area (number 1 in Figure 3) has no impact in the wave generation and, therefore, has the biggest cells to reduce as much as possible the computational

cost. The area below the extinction system (number 2 in Figure 3) has the same cell size than the air area due to its negligible influence in the experiments. The "deep-water" area is the section below the free surface that always contains water (number 3 in Figure 3). This area has smaller cells than the air area because it is affected by the particles' movement in the waves but is bigger than the free surface and extinction areas also to reduce the computational cost of the simulations. These last areas (number 4 and 5 in Figure 3) have a cell size that depends on the wave generated. The height of the cells is calculated by dividing the wave height (*H* [mm]) by 20, with this being the biggest cell size that ensures the good definition of the wave. The length of each cell is calculated by having an aspect ratio (AR) of 4, which is the relation between the length and height of the cell:

$$AR = \Delta_x/\Delta_y. \tag{3}$$

In the walls where an important interaction with the fluid exists, the paddle and extinction system, prism layers are introduced to have a better definition of the interaction. The total height of the free surface area ($FS_H$ [mm]) is calculated depending on the wave height too, with the total height of this area being:

$$FS_H = H + 20. \tag{4}$$

All the boundaries in the 2-D meshes have a non-slip condition and the bottom and the top have a morphing boundary plane constraint to reproduce a realistic movement of the paddle. In the 3-D mesh, the walls satisfy the non-slip condition except the lateral ones, with a slip condition to delete the friction influence of the walls in the wave generation and propagation. This allows a proper comparison with the experimental flume, in which this friction can be neglected due to its much larger width.

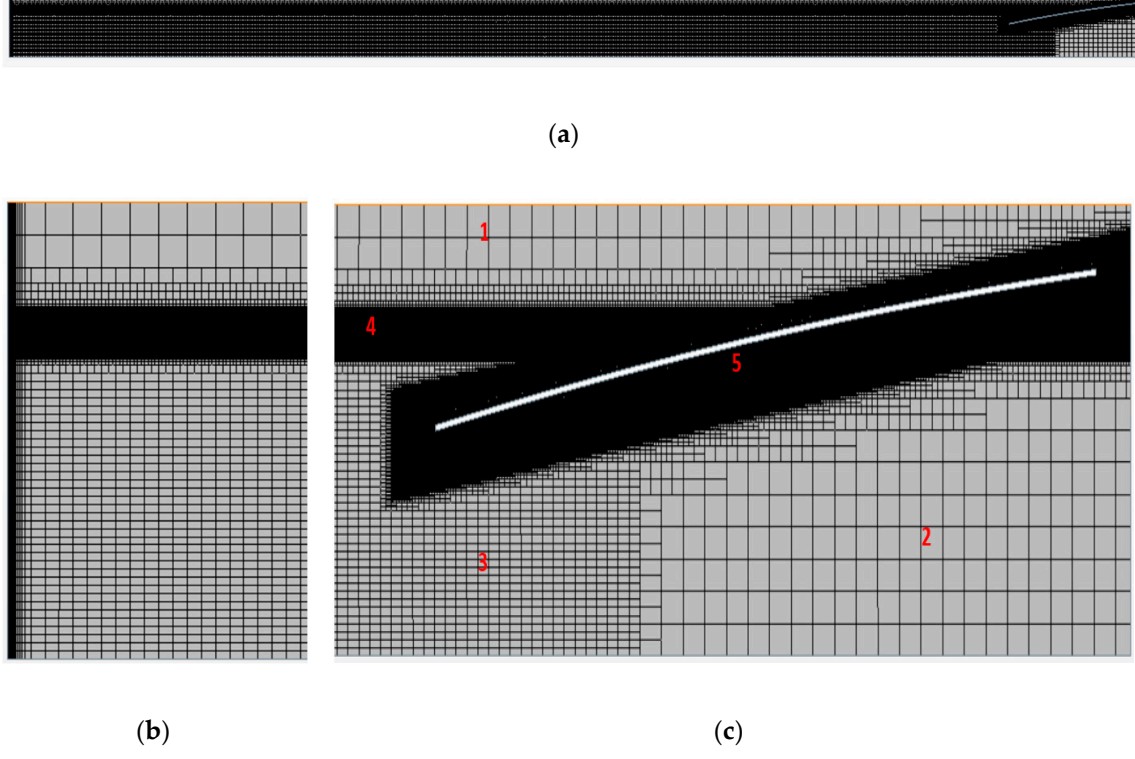

(**a**)

(**b**)                                          (**c**)

**Figure 3.** (**a**) Mesh corresponding to a depth of 50 cm, (**b**) focus on the prism layers around the generation paddle. (**c**) The different areas of the mesh: (1) air, (2) below the paddle, (3) "deep-water", (4) free surface, and (5) extinction system.

2.2.2. Reynolds-Averaged Navier–Stokes Equations (RANS)

Every Newtonian and continuum flow can be described using the Navier–Stokes equations. The turbulence is represented by the instantaneous values of the flow properties [8]. The equations for turbulence fluctuation are obtained by Reynolds de-composition, which substitutes instantaneous velocity and pressure fields for a mean value and a fluctuating component [27]. These components are substituted into the non-dimensional continuity and momentum equations using the time averaging:

$$\rho \bar{u}_j (\partial \bar{u}_i / \partial x_j) = \rho f_i + (\partial / \partial x_j)\ [-p \partial_{ij} + \mu((\partial \bar{u}_i / \partial x_j) + (\partial \bar{u}_j / \partial x_i)) - \rho u_i' u_j'], \tag{5}$$

where $\bar{u}_i$ is the time-averaged velocity, $p$ is the pressure field, $\rho$ is the density of the effective flow, $\mu$ is the viscosity, and $-\rho u_i' u_j'$ is the Reynolds operator.

2.2.3. Volume of Fluid Method

The VOF method presented by Hirt and Nichols [9] determines the percentage of the volume of a cell that is occupied by a certain fluid:

$$\alpha_i = V_i / V. \tag{6}$$

In this study, two fluids are used, water and air, and the free surface data are determined according to the volume fraction of the water, which is 1 if the cell is full of water and 0 if there is no water in the cell. For those cells where the free surface appears, the volume fraction of the water takes a value between 0 and 1, and this determines the amount of water that is in that cell [9]. The orientation of the free surface is perpendicular to the gradient of the volume fraction that the cell has in relation with its adjoining cells. The physical properties of each fluid are obtained from the fluids and their volume fraction as:

$$\rho = \Sigma_i\ (\alpha_i \rho_i), \tag{7}$$

$$\mu = \Sigma_i\ (\alpha_i \mu_i), \tag{8}$$

where $\rho_i$ is the density of each fluid and $\mu_i$ is their viscosity. The general conservation equation that describes the transport of volume fractions can be defined as:

$$(\partial / \partial t) \int_v (\alpha_i dV) + \int_s (\alpha_i\ (v - v_g) da) = \int_v (S_{\alpha i} - [(\alpha_i / \rho_i)(D\rho_i / Dt)] dV), \tag{9}$$

where $v$ is the velocity of the fluid, $v_g$ is the grid velocity, and $S_{\alpha i}$ is the source of the *i*th phase. It is important to highlight that, in our case, the source term is 0.

## 3. Aims and Methodology

The aim of the present work was to characterize different scenarios of wave generation to be fully reproduced inside a wave flume, aiming to be able to properly adjust the key parameters of the turbulence models to the real behavior of the numerical models, depending on the turbulence model that best adjusts to each particular case study. The methodology followed in this research can be divided in 4 different stages: first, the experiments were theoretically specified to ensure their possible reproduction by the wavemaker of the EWF. Thereafter, the planned experiments were carried out and the data obtained were processed in MATLAB, and compared with the theoretical values given by the wave theory. Next, the simulations were run to validate the numerical model by comparison with the experimental values. Finally, a comparison of experimental and numerical results when studying reflection accomplished all the objectives of this study.

*3.1. Experimental Procedures*

All the waves were generated following the same methodology. First, the position of the paddle was specified (amplitude) using the ASDA-Soft Software. After this, the period was specified,

which determines the time interval needed for each complete movement of the paddle. Since the wave period is the same as the period of the paddle, the theoretical value of the wavelength ($\lambda$) was calculated with the dispersion Equation (10), which relates the depth ($h$), period ($T$), and wavelength ($\lambda$). Then, the velocity propagation of the wave, ($c$ [m/s]), was calculated by (11) in order to determine the time, $t_E$, for which the wave will travel forward and backward along the EWF, until it reaches the position of the last probe (x$_3$) again:

$$\lambda = (gT^2/2\pi) \tanh(2\pi h/\lambda), \tag{10}$$

$$c = \lambda/T, \tag{11}$$

$$t_E = (2L_{Tank} + x_3)/c. \tag{12}$$

Each sensor acquired all the data corresponding to the free surface displacement ($\eta_i$ [m]). When the experiment in the EWF ended, the paddle was situated at 0.5 m from the wall to ensure the same starting point for every experiment.

### 3.2. Data Processing by MATLAB

Once the simulations and the experiments were finished, the data were analyzed by MATLAB to obtain the resulting parameters ($H$, $T$, and $\lambda$) of the incident wave. The results of the EWF were compared with the theoretical values to ensure the correct generation of the wave. Then, these results were used to calculate the corresponding reflection coefficient.

For the analysis of both the experimental or numerical test, the time domain was divided in three stages, depending on the characteristics of the wave: (i) the initial transitory when the wave is developing, (ii) the time interval in which the wave travels along the flume without any influence of reflected waves, and (iii) when the reflection affects the free surface displacement of the incoming incident wave. Therefore, the time interval corresponding to each stage must be defined. As an example, the $t_R$ corresponds to the instant when the reflection phenomenon is first measured on each probe:

$$t_R = (2L_{Tank} - x_i)/c, \tag{13}$$

where $L_{tank}$ is the length of the tank and x$_i$ is the position of each sensor. Then, the re-reflection time was calculated with Equation (12) for the position of each probe, respectively. To obtain the parameters of the waves, a specific height condition was created in MATLAB that disregards the data corresponding to the interval in which the wave is still developing. This condition was applied until the height was, at least, 70% of the maximum height before reflection appears. This time value was defined as the start time ($t_S$ [s]). In each analysis, the time before $t_S$ and after $t_E$ were deleted. Then, each run was divided into incident wave and reflection analysis.

For the analysis of the incident wave, the data between $t_S$ to $t_R$ were selected and fitted according to the corresponding wave theory, first (linear, Airy's theory) or second (non-linear, Stokes' theory) order, to obtain the experimental and numerical parameters of that wave ($H$, $T$, and $\lambda$). The values obtained from this fitting were compared to the theoretical values. It is important to highlight that the fitting avoids the variations from the theory that appear in the free surface displacement data.

Finally, the reflection coefficient ($K$ [-]) was calculated with the method developed by Mansard and Funke [35]. This method uses three wave probes to obtain $H$ at each position, and three phase-shift to calculate the amplitude value of both the incident ($A_I$) and the reflected wave ($A_R$). Then, the reflection coefficient is obtained:

$$K = A_R/A_I. \tag{14}$$

### 3.3. Numerical

Based on the experiments theoretically defined, a wave selection was carried out to reproduce numerically waves of different properties for the validation of the models. Then, linear and non-linear

waves at the intermediate and deep-water regions were simulated. These simulations were created following the descriptions of Section 2.2. Then, the mesh was converted to 2-D for the *k-ε* or *k-ω* models to reduce the computational cost, taking advantage of the 2-D nature of the EWF. For the LES model, STAR CCM+ does not allow a 2-D mesh, so a 5-mm-wide mesh was created to not create an excessive increase in the number of cells. Perpendicular planes were defined to replicate the resistive-type probes of the EWF and measure the free surface displacement at the exact position than in the EWF, for which a user-generated function was imposed to determine the position of the free surface. This function uses the volume fraction of the water that can be defined as:

$$\eta_{\text{sim}} = (\text{VOF}_{\text{water}} \cdot 700) - h_{study}, \tag{15}$$

where $\eta_{\text{sim}}$ is the position of the free surface in mm, $\text{VOF}_{\text{water}}$ is the mean volume fraction of water measure in the study plane, 700 is the total height of the NWF in mm, and $h_{study}$ is the depth of study in mm. Thus, the value of $\eta_{sim}$ at the beginning of each simulation was 0.

### 3.4. Results Comparison

The results were compared for all the turbulence models and for both the theoretically and experimentally obtained results. This comparison was carried out to study the feasibility of the turbulence models for the wave generation, which was the main objective of this research. In order to minimize the variables, the turbulence models were compared at a constant cell size.

## 4. Turbulence Models

To have a better definition of the turbulence in the RANS equations, different models have been created throughout the years. Normally, they are ordered with the number of equations that they used, such as zero-equation, one-equation, or two-equation models. In this article, we used two variations of the two-equation models of *k-ε* and *k-ω*, apart from the large-eddy simulations.

### 4.1. The Low Reynolds k-ε Model

The standard *k-ε* low-Re model has the same coefficients as the standard *k-ε* model; however, it introduced some damping functions to enable its application in the viscous affected regions near-wall [36], making it more compatible with the law of the wall. However, it does not solve the problem with adverse pressure gradients. In this model, the kinematic eddy viscosity can be defined as:

$$\nu_{\text{t}} = C_\mu f_\mu (k^2/\varepsilon), \tag{16}$$

where $C_\mu$ is a model constant, $f_\mu$ is a damping function, $k$ is the turbulent kinetic energy, and $\varepsilon$ is the turbulence dissipation rate. The mathematical modelling can be written in a boundary layer form:

$$\bar{u}(\partial k/\partial x) + v(\partial k/\partial y) = (\partial/\partial y)[(\nu + (\nu_t/\sigma_k))\,(\partial k/\partial y)] + \nu_t(\partial u/\partial y)^2 - \varepsilon$$
$$\bar{u}(\partial \varepsilon/\partial x) + v(\partial \varepsilon/\partial y) = (\partial/\partial y)[(\nu + (\nu_t/\sigma_k))\,(\partial \varepsilon/\partial y)] + C_{\varepsilon 1}f_1(\varepsilon/k)\nu_t(\partial u/\partial y)^2 - C_{\varepsilon 2}f_2(\varepsilon^2/k) + E. \tag{17}$$

In the standard model, the turbulence kinetic energy is defined as:

$$\partial k/\partial t + \bar{u}_j(\partial k/\partial x_j) = \partial/\partial x_j\,[\nu + \nu_t/\sigma_k\,\partial k/\partial x_j] - \varepsilon + \tau_{ij}\partial \bar{u}_i/\partial x_j. \tag{18}$$

Turbulence dissipation rate ($\varepsilon$) equation:

$$\partial \varepsilon/\partial t + \bar{u}_j(\partial k/\partial x_j) = \partial/\partial x_j\,[\nu + \nu_t/\sigma_k\,\partial \varepsilon/\partial x_j] - C_{\varepsilon 1}(\varepsilon/k)\tau_{ij}(\partial \bar{u}_i/\partial x_j) - C_{\varepsilon 2}(\varepsilon^2/k), \tag{19}$$

where the $\sigma_k = 1$ and $\sigma_\varepsilon = 1.3$ are the Prandtl numbers for $k$ and $\varepsilon$. The damping functions in the standard *k-ε* low-Re are $f_2$:

$$f_2 = 1 - C \cdot \exp(-\text{Re}_t^2), \tag{20}$$

and $f_\mu$:

$$f_\mu = 1 - \exp[-(C_{d0}\sqrt{(Re_d)} + C_{d1}Re_d + C_{d2}Re_d{}^2)]. \tag{21}$$

## 4.2. The Shear Stress Transport (SST) Model

First presented by Kolmogorov [37], the most important development of the *k-ω* model was presented by Wilcox and is the one that the program Star-CCM+ uses as a standard. This model is a two-equation model that has clear advantages in comparison with *k-ε*, achieving higher accuracies in boundary layers with adverse pressure gradients and in shear and separated flows. This model defines the kinematic viscosity as:

$$\nu_t = (k/\omega), \tag{22}$$

where *k* is the turbulence kinetic energy and *ω* is the specific dissipation rate. The turbulence kinetic energy (*k*) is defined as:

$$\partial k/\partial t + \bar{u}_j(\partial k/\partial x_j) = \partial/\partial x_j[(\nu + \sigma^* k/\omega)\,\partial k/\partial x_j]\beta^*\,k\omega + \tau_{ij}\,(\partial \bar{u}_i/\partial x_j). \tag{23}$$

The specific dissipation rate (*ω*) equation is:

$$\partial \omega/\partial t + \bar{u}_j(\partial \omega/\partial x_j) = \partial/\partial x_j\,[(\nu + \sigma k/\omega)\,\partial \omega/\partial x_j] - \beta\,k\omega^2 + (\sigma_d/\omega)(\partial k/\partial x_j)(\partial \omega/\partial x_j) + a\,(\omega/k)\tau_{ij}\,(\partial \bar{u}_i/\partial x_j). \tag{24}$$

The SST model developed by Menter combines advantages from both *k-ε* and *k-ω*. Its complete formulation is in [13], where the author shows that the model works with small variances in the shear-stress and improvements for the predictive accuracy for industrial application, reducing the influence of the user-generated grid.

## 4.3. Large-Eddy Simulation (LES)

In the LES approach, the large-scale turbulence is fully resolved while the sub-grid turbulence is modelled. In comparison with RANS approach, only the small isotropic turbulent scales are modelled. It is extremely useful for applications with high Reynolds numbers. The filtered Navier–Stokes equations of LES can be defined as [38]:

$$\begin{aligned}\partial\rho/\partial t + \partial\rho u_i/\partial x_i &= 0\\\partial\rho u_i/\partial t + \partial\rho u_i u_j/\partial x_j + \partial p/\partial x_i - \partial\sigma_{ij}/\partial x_j &\approx -\partial\tau_{ij}/\partial x_j\\\partial\rho E/\partial t + \partial(\rho E + p)u_j/\partial x_j - \partial u_i\sigma_{ij}/\partial x_i - \partial q_j/\partial x_j &\approx -1/\gamma - 1\,\partial(pu_j - pu_j)/\partial x_j - u_j(\partial\tau_{ij}/\partial x_j).\end{aligned} \tag{25}$$

However, a subgrid model is necessary to have a good definition. For this, the one developed by Smagorinsky was used. This model uses a mixing-length hypothesis to model the subgrid-scale stresses [36].

It is based on a sufficiently high Reynolds number, which ensures the energy transfer from large to small scales, which are responsible for dissipation. It predicts that eddy viscosity reaches its highest values in areas, for example, solid boundaries, where a strong shear appears. The model provides the mixing-length type formula for the subgrid viscosity:

$$\mu_t = \rho\Delta^2 S, \tag{26}$$

where *ρ* is the density, *Δ* is the length scale, and *S* is given resolving the velocity field:

$$\Delta = f_v\,C_s\,V^{1/3}, \tag{27}$$

where $C_S$ is the model coefficient and $f_v$ is the Van Driest damping function, which in the software can be computed as:

$$f_v = 1 - \exp(-y^+/A), \tag{28}$$

where *A* is another model constant and wall $y^+$ is the dimensionless wall distance.

## 5. Results and Discussion

Six different waves were used in this study to compare the turbulence models, the defining parameters of which are shown in Table 1. These theoretical values were obtained by combining the Le Méhauté chart [39] and the dispersion Equation (10). The selected wave heights and periods, combined with a wide range of depths in the EWF, from 0.3 to 0.5 m, were selected in order to represent the different waves that appear in the Biscay Marine Energy Platform (BIMEP) [40,41].

**Table 1.** Theoretical values of the studied waves.

|        | h (m) | *H* (mm) | *T* (s) | λ (m) | c (m/s) |
|--------|-------|----------|---------|-------|---------|
| Wave 1 | 0.3   | 40       | 1.25    | 1.88  | 1.54    |
| Wave 2 | 0.3   | 30       | 1.42    | 2.20  | 1.54    |
| Wave 3 | 0.4   | 39       | 0.81    | 1.01  | 1.24    |
| Wave 4 | 0.4   | 64       | 1.27    | 2.11  | 1.65    |
| Wave 5 | 0.5   | 29       | 1.20    | 2.06  | 1.71    |
| Wave 6 | 0.5   | 60       | 1.42    | 2.64  | 1.85    |

The fittings of the free surface displacement signals were done using the incident wave interval, which corresponds to all the data between the $t_S$ and $t_R$ time values. This fitting, shown in Figure 4, matches almost perfectly with the free surface variation data obtained from the experimental runs. This adjustment constantly achieves a quality of overlapping over 95% in the 3 probes for every wave.

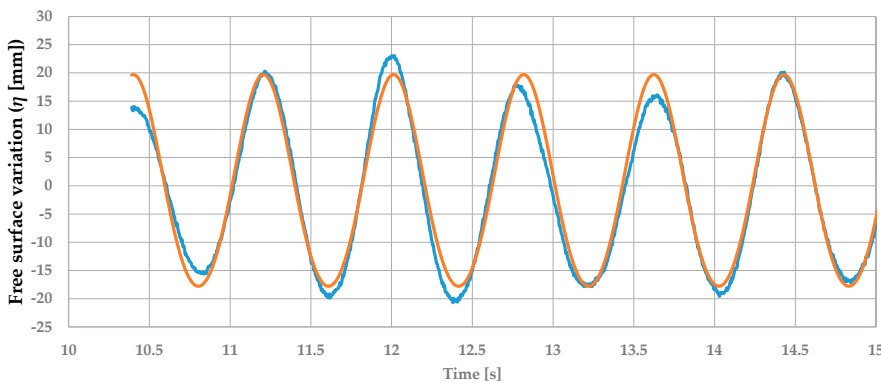

**Figure 4.** Fitting of the free surface displacement for the wave 3: fitting (orange) and experimental (blue) measured by the first probe of the EWF.

The parameters of the wave were obtained by calculating the average of the parameters of the fittings obtained from the signals of the three probes. Then, the values were compared with the theoretical ones. The results and the corresponding relative errors in relation to the theoretical values are shown in Table 2.

**Table 2.** Experimental results and relative errors of the incident wave with respect to the theoretically intended values.

|        | *H* (mm) | *T* (s) | λ (m) | c (m/s) | Error *H* (%) | Error *T* (%) | Error λ (%) | Error c (%) |
|--------|----------|---------|-------|---------|---------------|---------------|-------------|-------------|
| Wave 1 | 42.07    | 1.25    | 1.78  | 1.42    | 5.41          | −0.63         | −7.08       | −6.48       |
| Wave 2 | 30.04    | 1.41    | 2.10  | 1.49    | 0.14          | 1.19          | 4.59        | 3.40        |
| Wave 3 | 36.89    | 0.79    | 0.99  | 1.24    | 5.39          | 1.36          | 1.47        | 0.10        |
| Wave 4 | 66.29    | 1.27    | 2.11  | 1.65    | 3.58          | 0.15          | 0.15        | 0.01        |
| Wave 5 | 28.34    | 1.18    | 2.02  | 1.71    | 2.26          | 1.80          | 1.83        | 0.03        |
| Wave 6 | 56.50    | 1.41    | 2.68  | 1.90    | 5.82          | 1.23          | 1.78        | 3.06        |

Focusing on the relative errors, it can be concluded that the piston-type wavemaker appropriately generates the waves according to the wave theory. Only for the wave 1 were relative errors higher than 6% measured.

The computational turbulence models were then compared, following a similar methodology. In this case, the validation of the models was carried out using the experimental results obtained in the EWF as a reference. This step was essential to compare the quality of the models and its computational cost.

The $k$-$\varepsilon$ turbulence model is the one that was first used in this and previous studies [5]. However, a better definition of the turbulent kinetic energy and the turbulence dissipation rate was obtained to improve the accuracy of the simulations. These values were modified to $1 \times 10^{-5}$ for the turbulent kinetic energy and $1 \times 10^{-4}$ for the turbulence dissipation rate. These changes help to improve the fitting between experimental and numerical values by defining the crests of the waves better, although troughs are yet to be better defined.

The wave parameters obtained from the simulations using the $k$-$\varepsilon$ turbulence model can be seen in Table 3. A limit of 10% for the wave height and 2% for the period were selected to validate the models. Although the errors are low, always below 2% in period, a better definition of the paddle movement is needed because errors become higher when higher waves are generated. It is important to highlight the satisfactory validation of the equation that the moving wall follows. Next, the wavelength was calculated with the phase-shift of the signals between probes and finally the wave propagation velocity, $c$. It is worth mentioning that the calculation of the wavelength through the phase-shift introduces a cumulative error from the signal fitting, so it will be further studied for its improvement.

**Table 3.** Results obtained from the simulations carried out using the $k$-$\varepsilon$ turbulence model and relative errors with respect to the experimental values of the incident wave.

|  | $H$ (mm) | $T$ (s) | $\lambda$ (m) | $c$ (m/s) | Error $H$ (%) | Error $T$ (%) | Error $\lambda$ (%) | Error $c$ (%) |
|---|---|---|---|---|---|---|---|---|
| Wave 1 | 39.15 | 1.25 | 1.84 | 1.46 | −6.96 | −0.09 | 3.30 | 2.54 |
| Wave 2 | 29.42 | 1.42 | 2.20 | 1.54 | −2.08 | 0.64 | 4.48 | 3.19 |
| Wave 3 | 36.90 | 0.80 | 1.00 | 1.24 | −5.39 | −1.36 | −1.24 | −0.69 |
| Wave 4 | 63.65 | 1.27 | 2.14 | 1.67 | −3.99 | −0.70 | 1.06 | 0.86 |
| Wave 5 | 28.31 | 1.20 | 2.18 | 1.81 | −0.12 | 1.25 | 7.47 | 5.76 |
| Wave 6 | 60.56 | 1.42 | 2.68 | 1.87 | 7.17 | 0.69 | −0.37 | −1.97 |

The results obtained for the $k$-$\varepsilon$ model were close to the experimental ones, so that the same mesh was used for the simulations carried out with the SST turbulence model. The value for the turbulent kinetic energy was the same as for the $k$-$\varepsilon$ model, as well as the value of the specific dissipation rate ($\omega$). The similar behavior of both two-equation models reinforces the idea of the use of each model for the problem that best suits each of them. The tendency in the relative errors is the same in both models, as it can be seen in Table 4. However, the SST model has a worst definition in the measurements of the wavelength and celerity, while its definition of the wave height has marginal improvements when comparing it to the $k$-$\varepsilon$ model.

**Table 4.** Results obtained from the simulations carried out using the SST turbulence model and relative errors with respect to the experimental values of the incident wave.

|  | $H$ (mm) | $T$ (s) | $\lambda$ (m) | $c$ (m/s) | Error $H$ (%) | Error $T$ (%) | Error $\lambda$ (%) | Error $c$ (%) |
|---|---|---|---|---|---|---|---|---|
| Wave 1 | 39.28 | 1.25 | 1.88 | 1.49 | −6.65 | −0.09 | 5.54 | 4.65 |
| Wave 2 | 29.52 | 1.42 | 2.22 | 1.55 | −1.74 | 0.64 | 5.43 | 3.86 |
| Wave 3 | 36.89 | 0.80 | 1.00 | 1.24 | −5.39 | −1.36 | −1.24 | −0.69 |
| Wave 4 | 63.12 | 1.28 | 2.03 | 1.59 | −4.79 | 0.09 | −4.13 | −3.97 |
| Wave 5 | 28.29 | 1.19 | 1.97 | 1.65 | −0.19 | 0.40 | −2.88 | −3.59 |
| Wave 6 | 59.64 | 1.42 | 2.84 | 1.99 | 5.54 | 0.69 | 5.58 | 4.33 |

As aforementioned, the LES model does not allow simulations in a 2-D mesh to be carried out using Star CCM+. Despite using a 3-D mesh, the same rules of cell size were applied, having the same height and length as the 2-D meshes. The model uses the coefficients $C_S = 0.1$, $C_t = 3.5$, $A = 25$, and the von Karman constant $\kappa = 0.41$. Table 5 shows the values and relative errors of each simulation using this LES model. The adjustments of wave height are worse compared to the two-equation models, but the definition of the wavelength is in the same error interval as the two other models. Because of this, it can be concluded that this model reproduces significantly well the experimentally generated waves.

**Table 5.** Results obtained from the simulations carried out using the LES turbulence model and relative errors with respect to the experimental values of the incident wave.

|  | $H$ (mm) | $T$ (s) | $\lambda$ (m) | $c$ (m/s) | Error $H$ (%) | Error $T$ (%) | Error $\lambda$ (%) | Error $c$ (%) |
|---|---|---|---|---|---|---|---|---|
| Wave 1 | 37.73 | 1.26 | 1.88 | 1.49 | −10.34 | 0.55 | 5.51 | 4.93 |
| Wave 2 | 28.31 | 1.43 | 2.21 | 1.55 | −5.76 | 1.10 | 5.01 | 3.87 |
| Wave 3 | 36.89 | 0.81 | 1.01 | 1.24 | −5.39 | −0.06 | −0.49 | −0.43 |
| Wave 4 | 62.44 | 1.28 | 2.03 | 1.59 | −5.82 | 0.27 | −3.91 | −4.16 |
| Wave 5 | 28.99 | 1.20 | 1.97 | 1.65 | 2.27 | 1.08 | −2.84 | −3.88 |
| Wave 6 | 58.91 | 1.42 | 2.84 | 2.00 | 4.25 | 0.84 | 5.57 | 4.68 |

A comparison of the free surface displacement was carried out between experiments and simulations because of the low errors measured. This way allows a direct comparison between incident and reflected waves and observation of whether the extinction system affects the free surface displacement. Figure 5 shows these comparisons between the experimental and the three turbulence models used in the first probe. The fitting is significantly good throughout all the experiment, generating a similar energy dissipation by the extinction system, resulting in a similar free surface displacement at all three points. Figure 5 shows a good definition of the wave crests in all the models, but they fall short on the definition of the troughs, for which the LES models achieve a more constant definition of them.

Once it was demonstrated that the free surface displacement fits properly with the $k$-$\varepsilon$, SST and LES turbulent models for the incident wave, a statistical comparison of the adjustments of the free surface displacement when reflection appears was developed in order to determine the influence of the energy dissipation process and its turbulence modelling. This comparison focused on the wave height and wave period, as shown in Table 6. From this table, it can be concluded that the $k$-$\varepsilon$ turbulent model obtained promising results, but high error in the height was measured for the wave with the shortest period.

**Table 6.** Comparison of the wave height and period of the resultant wave between the experimental and $k$-$\varepsilon$ model, and the error between them.

|  | $H_{exp}$ (mm) | $T_{exp}$ (s) | $H_{k\text{-}\varepsilon}$ (mm) | $T_{k\text{-}\varepsilon}$ (s) | Error $H$ (%) | Error $T$ (%) |
|---|---|---|---|---|---|---|
| Wave 1 | 40.34 | 1.26 | 38.83 | 1.25 | −3.73 | −0.79 |
| Wave 2 | 31.02 | 1.45 | 31.72 | 1.42 | 2.26 | −1.66 |
| Wave 3 | 37.13 | 0.81 | 34.46 | 0.81 | −7.19 | 0.41 |
| Wave 4 | 65.27 | 1.25 | 63.76 | 1.25 | −2.31 | 0.50 |
| Wave 5 | 28.50 | 1.18 | 28.05 | 1.20 | −1.61 | 1.88 |
| Wave 6 | 56.78 | 1.43 | 59.42 | 1.41 | 4.66 | −1.50 |

Similar results were obtained with the SST model. While the definition of the wave periods is almost identical, a slight improvement is made in the wave height definition. The results can be seen in Table 7, where wave heights do not surpass a 7% error, while wave periods are always maintained at lower than 2%. This can be due to the improvements introduced by Menter, where the behavior of

the adverse pressure gradients is improved and thus, a better definition of the troughs is obtained, reaching, in some cases, a better definition.

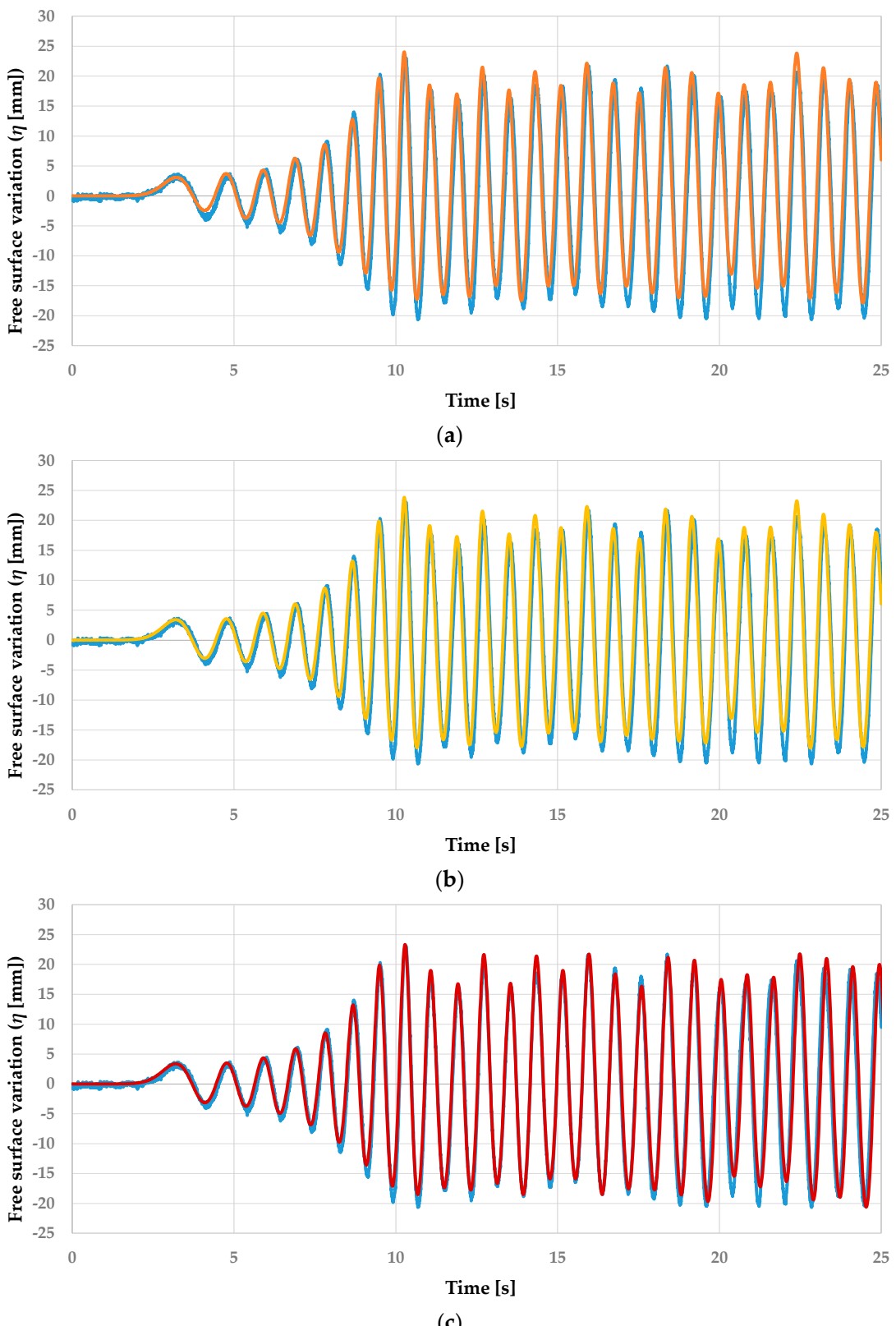

**Figure 5.** Comparison of the free surface displacement for the turbulence model of (**a**) *k-ε*, (**b**) SST, and (**c**) LES with the experimental data (blue) obtained in the EWF for the wave 3.

**Table 7.** Comparison of the wave height and period of the resultant wave between the experimental and SST model, and the error between them.

| | $H_{exp}$ (mm) | $T_{exp}$ (s) | $H_{SST}$ (mm) | $T_{SST}$ (s) | Error $H$ (%) | Error $T$ (%) |
|---|---|---|---|---|---|---|
| Wave 1 | 40.34 | 1.26 | 38.69 | 1.25 | −4.08 | −0.78 |
| Wave 2 | 31.02 | 1.45 | 31.59 | 1.42 | 1.85 | −1.70 |
| Wave 3 | 37.13 | 0.81 | 34.80 | 0.81 | −6.28 | 0.30 |
| Wave 4 | 65.27 | 1.25 | 63.76 | 1.25 | −2.31 | 0.50 |
| Wave 5 | 28.50 | 1.18 | 28.16 | 1.20 | −1.21 | 1.87 |
| Wave 6 | 56.78 | 1.43 | 59.40 | 1.41 | 4.61 | −1.45 |

Finally, the LES results are shown in Table 8. Although a better definition of the waves is achieved with this model in deeper depths, it shows limitations when working at smaller depths. However, the tendency of the two other turbulence models when defining the wave period is constant and a better definition of the troughs is reached.

**Table 8.** Comparison of the wave height and period of the resultant wave between the experimental and LES model, and the error between them.

| | $H_{exp}$ (mm) | $T_{exp}$ (s) | $H_{LES}$ (mm) | $T_{LES}$ (s) | Error $H$ (%) | Error $T$ (%) |
|---|---|---|---|---|---|---|
| Wave 1 | 40.34 | 1.26 | 37.10 | 1.26 | −8.03 | 0.05 |
| Wave 2 | 31.02 | 1.45 | 30.29 | 1.42 | −2.35 | −1.66 |
| Wave 3 | 37.13 | 0.81 | 36.47 | 0.81 | −1.76 | 0.68 |
| Wave 4 | 65.27 | 1.25 | 64.56 | 1.25 | −1.08 | 0.54 |
| Wave 5 | 28.50 | 1.18 | 28.82 | 1.20 | 1.12 | 1.89 |
| Wave 6 | 56.78 | 1.43 | 59.43 | 1.41 | 4.66 | −1.46 |

Finally, the reflection of the extinction system was studied to obtain more data for a proper comparison of the behavior of the turbulence models. The wave breaking, which is related to reflection, is the most turbulent process seen in the flume and therefore, it can result in higher differences between the turbulent models under investigation. Besides, the reflection coefficients allow comparison of the amplitudes of the incident and reflected waves, which confirms both fittings made to the free surface displacement signal.

The reflection method developed by Mansard and Funke uses both heights and phase-shifts to calculate the amplitudes of the incident and the reflected waves. The reflection coefficient is related to the dissipated energy produced in the extinction system by the turbulent process that occurs. Because of this, the study of the reflection phenomenon has great importance when defining the turbulence model and the NWF. In Figure 6, a comparison of the reflection coefficients measured for all the waves, both numerical and experimental, is shown. Despite the different values measured for the coefficients, it is important to highlight that the errors using this method are magnified by the errors measured in the incident and reflected waves.

A direct comparison of the incident and reflected amplitudes is shown in Figure 7 using the method developed by Mansard and Funke. This provides better results in the study of the amplitude of the incident wave when treating the simulations. This can be related to the high sensitivity of the resistive probes of the EWF, which introduce noise in the signal and makes the measurements in some waves difficult. Further research will be needed to improve the implementation of the method in the experiments. Besides, the small variations in the reflected amplitude are magnified when dividing it by the amplitude of the incident wave. For example, for the wave 1, the incident amplitude of the experimental signal is significantly smaller; however, a marginally smaller amplitude of the reflected wave generates a much smaller reflection coefficient.

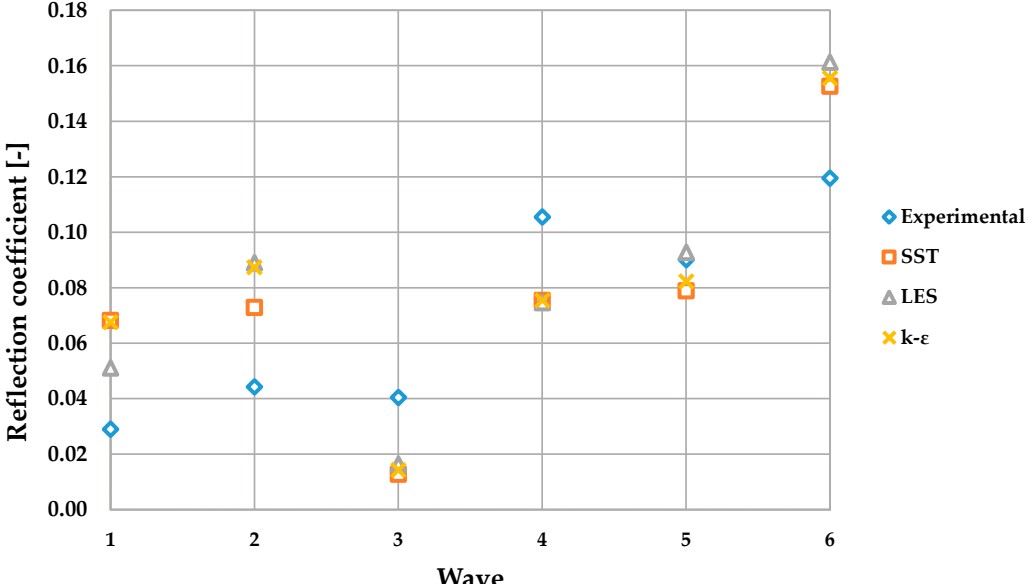

**Figure 6.** Reflection coefficients measured for each wave in the experimental and numerical runs.

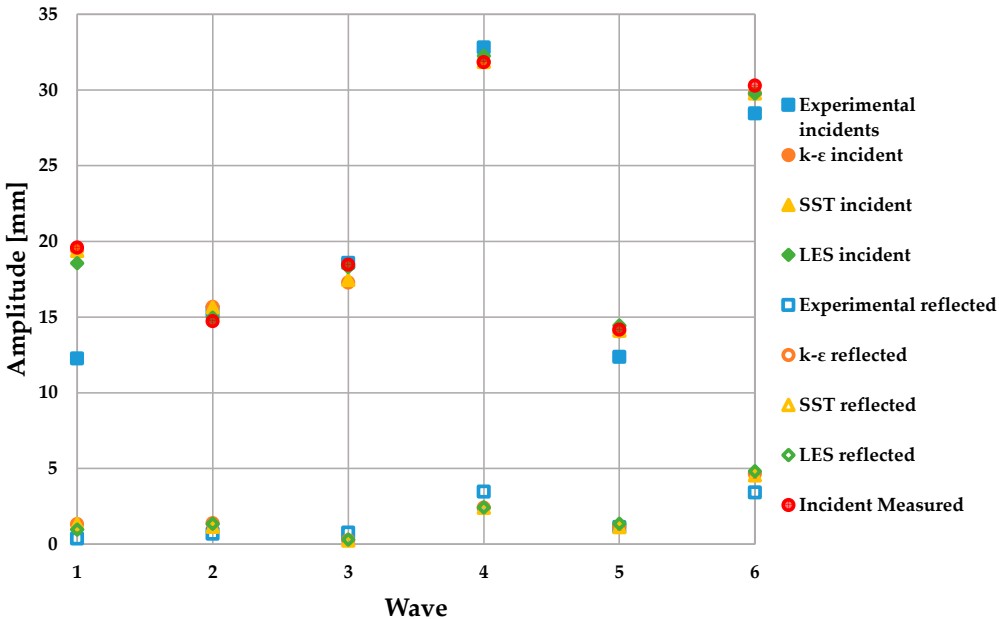

**Figure 7.** Comparison of the incident and reflected amplitudes of the waves by the reflection method developed by Mansard and Funke for the experimental and numerical runs.

## 6. Conclusions

In this study, three different turbulence models were compared ($k$-$\varepsilon$, $k$-$\omega$, and LES) and validated using the EWF of the University of the Basque Country. To the knowledge of the authors, hitherto, this is the first work comparing the three turbulence models to determine which of them reproduces the reflection phenomenon studied in the NWF better. The three models showed good correlations when measuring free surface variation in comparison with the experimental data measured in the laboratory.

A more specific study was made in wave generation, testing if the movement of the wavemaker of the NWF follows the wavemaker theory. The performance of the moving wall showed good results, validating the movement equations that were calculated, and accurately matching the wave period and wave height.

The fitting carried out using MATLAB for the experimental and numerical free surface displacement allowed the parameters of the waves to be obtained and compared with each other, as well as the values theoretically calculated.

The main contribution of this research related to the reflection coefficients that were calculated to compare properly the three turbulence models between each other, and with the experimentally obtained values. The reflection produced by the extinction system was studied in three different ways. The first one, in which a direct comparison of the free surface variation was made, showed good agreement between the experimental and the numerical signals, despite slight disparities in the troughs. The reflection coefficient study showed promising results, although clear differences were seen in them. The results for the simulations showed strong similarity when comparing the incident wave amplitude measured using the Mansard and Funke method with the value measured with the probes. However, the experimental signals showed significant differences in some waves. This could be due to the noise and will be studied in future articles.

Finally, the three studied turbulence models showed a good performance throughout all the time of the research, although they did not achieve a perfect matching with the experimental results. The aim of the article, the use of three different turbulence models to simulate the behavior of the wave flume, was achieved, creating a strong base for future studies, which will be the focus in the fluid–structure interaction of floating breakwaters or wave energy converters, such as oscillating water columns.

**Author Contributions:** U.I. and L.G.-C. designed and run the experimental tests, while L.G.-C. did the numerical simulations. J.M.B. and G.A.E. are responsible of the analysis and validation of the experiments. All the authors have contributed to the writing and revision process of this article. All authors have read and agreed to the published version of the manuscript.

**Funding:** This research received no external funding.

**Acknowledgments:** The authors would like also to express their gratitude for the funding provided to the Research Groups of the UPV/EHU (GIU19/276) and the Basque Government (IT1314-19), as well as the support provided by the Joint Research Laboratory on Offshore Renewable Energy (JRL-ORE).

**Conflicts of Interest:** The authors declare no conflict of interest.

## Abbreviations

The following abbreviations are used in this manuscript:

| | |
|---|---|
| $A$ | Amplitude |
| $A_I$ | Amplitude of the incident wave |
| $A_R$ | Amplitude of the reflected wave |
| AR | Aspect Ratio |
| c | Wave celerity |
| CFD | Computational Fluid Dynamics |
| $C_\mu$ | $k$-$\varepsilon$ model constant |
| EWF | Experimental Wave Flume |
| FOWT | Floating Offshore Wind Turbine |
| $FS_H$ | Free Surface Height |
| $f_\mu$ | Damping function |
| $H$ | Wave Height |
| $K$ | Reflection Coefficient |
| $k$ | Kinetic energy |
| $L_{Tank}$ | Tank Length |
| LES | Large Eddy Simulation |
| NWT | Numerical Wave Tank |
| OWC | Oscillating Water Column |
| $p$ | Pressure |
| RANS | Reynolds-Average Navier Stokes |
| SST | Shear Stress Transport |
| u | Fluid velocity |

| | |
|---|---|
| V | Volume of the cell |
| $v$ | velocity |
| $v_g$ | Grid velocity |
| VOF | Volume of Fluid |
| $x_i$ | Position of the probe |
| $\alpha i$ | Volume Fraction of a fluid |
| $\Delta$ | Length Scale |
| $\varepsilon$ | Turbulence dissipation rate |
| $\eta$ | Free Surface Position |
| $\mu$ | Dynamic viscosity |
| $\nu_t$ | Kinematic viscosity |
| $\omega$ | Specific dissipation rate |
| $\sigma_\varepsilon$ | Prandtl constant for turbulence dissipation rate |
| $S_{ij}$ | Mean strain-rate tensor |
| $\sigma_k$ | Prandtl constant for kinematic energy |
| $T$ | Period |
| $t$ | time |
| $\tau_{ij}$ | Favre_averaged specific Reynolds-stress tensor |
| $\tau_{xy}$ | Favre-specific Reynolds shear stress |
| $\mu_T$ | Subgrid viscosity |
| NWF | Numerical Wave Flume |

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
