# Peer review of "Performance Assessment of Three Turbulence Models Validated through an Experimental Wave Flume under Different Scenarios of Wave Generation"

_jmse, doi:10.3390/jmse8110881_

Round 1

Reviewer 1 Report

In the present work, several turbulence models were considered for validation by using as a reference an experimental wave flume. The research line is easy to follow, and the results are solid enough. Definitely there is quality in this work and from my point of view this can be considered for publication as it is.

Reviewer 2 Report

Comments on manuscript “Performance assessment of three turbulence models validated through and experimental wave flume under different scenarios of wave generation”

In this study, different turbulence models to adequate them for future research such as oscillating water column devices behavior or breaking waves studies are compared. The study contains interested results that deserve to publish in Journal of Marine Science and Engineering. But a major revision of this manuscript is necessary.

Please find below the major criticisms:

What is the innovation of this study? If the innovation and the implementation of the present study were addressed in the revised manuscript, it can be reconsidered for publish in Journal of Marine Science and Engineering. You must refer in the conclusions the innovation and the implementation of the present study.

Small points:

  1. How you chose the height of the six different waves? Random?
  2. Until which percentage is acceptable between experimental and k-ε, SST and LES models?
  3. More details about software ASDA-Soft are necessary.

Round 2

Reviewer 2 Report

The authors made all the necessary changes that I mentioned.